# An antiferromagnetic spin phase change memory

Han Yan[1,7], Hongye Mao[1,7], Peixin Qin [1] ✉, Jinhua Wang[2], Haidong Liang [3], Xiaorong Zhou[1], Xiaoning Wang[1], Hongyu Chen[1], Ziang Meng[1], Li Liu[1], Guojian Zhao[1], Zhiyuan Duan[1], Zengwei Zhu [2], Bin Fang[4], Zhongming Zeng [4], Andrew A. Bettiol [3], Qinghua Zhang [5] ✉, Peizhe Tang [1,6] ✉, Chengbao Jiang [1] ✉ & Zhiqi Liu [1] ✉

The electrical outputs of single-layer antiferromagnetic memory devices relying on the anisotropic magnetoresistance effect are typically rather small at room temperature. Here we report a new type of antiferromagnetic memory based on the spin phase change in a Mn-Ir binary intermetallic thin film at a composition within the phase boundary between its collinear and noncollinear phases. Via a small piezoelectric strain, the spin structure of this composition-boundary metal is reversibly interconverted, leading to a large nonvolatile room-temperature resistance modulation that is two orders of magnitude greater than the anisotropic magnetoresistance effect for a metal, mimicking the well-established phase change memory from a quantum spin degree of freedom. In addition, this antiferromagnetic spin phase change memory exhibits remarkable time and temperature stabilities, and is robust in a magnetic field high up to 60 T.

For more than 70 years since its discovery by M. Louis Néel[1] in 1948, antiferromagnets have remained underutilized as key functional materials in spintronic devices. Instead, they have been primarily employed as pinning layers in devices such as spin valves because of their inherent net zero magnetic moments and ability to induce exchange bias with ferromagnetic materials. However, spintronic devices with antiferromagnets as key active elements have recently been developed and have attracted considerable attention owing to the absence of stray fields, ultrafast spin dynamics, and insensitivity to external magnetic fields[2–9], such as low-temperature spin-valve-like antiferromagnetic tunnel junctions[10], room-temperature antiferromagnetic memory resistors switched by thermal heating assisted magnetic field[11] or electrical-current-induced Néel spin-orbit torque[12],

and piezoelectric-strain-controlled antiferromagnetic memory devices[13–15]. In particular, piezoelectric strain excited by electric fields as an information writing method could be an ultralow-power mechanism due to the inhibition of Joule heating[16–18].

However, a longstanding bottleneck issue in single-layer antiferromagnetic memory devices is that their electrical outputs based on the traditional relativistic anisotropic magnetoresistance are exceedingly small, approximately 0.1% at room temperature[11,12,19]. The amplification of the output signals of an antiferromagnetic memory has caused many challenges. To overcome this issue, inspired by the well-established phase change memory, where the resistance change of several orders of magnitude is achieved by the interconversion of amorphous and crystalline phases upon electrical excitations, we

[1]School of Materials Science and Engineering, Beihang University, Beijing 100191, China. [2]Wuhan National High Magnetic Field Center, Huazhong University of Science and Technology, Wuhan 430074, China. [3]Centre for Ion Beam Applications (CIBA), Department of Physics, National University of Singapore, Singapore 117542, Singapore. [4]Key Laboratory of Multifunctional Nanomaterials and Smart Systems, Suzhou Institute of Nano-Tech and Nano-Bionics, Chinese Academy of Sciences, Suzhou 215123, China. [5]Beijing National Laboratory for Condensed Matter Physics, Institute of Physics, Chinese Academy of Sciences, Beijing 100190, China. [6]Max Planck Institute for the Structure and Dynamics of Matter, Center for Free Electron Laser Science, Hamburg 22761, Germany. [7]These authors contributed equally: Han Yan, Hongye Mao. ✉e-mail: qinpeixin@buaa.edu.cn; zqh@iphy.ac.cn; peizhet@buaa.edu.cn; jiangcb@buaa.edu.cn; zhiqi@buaa.edu.cn

propose an idea of an antiferromagnetic spin phase change memory, for example, when the spin structure of an antiferromagnet changes from collinear to noncollinear, the resistance of this material could be remarkably altered.

We applied our hypothesis of this spin phase change memory to the Mn-Ir binary intermetallic system; this system exhibits a high-temperature ($T_N \sim 1145$ K) collinear antiferromagnetic $L1_0$ MnIr phase when the Ir ratio is between 40% and 55%[20–25] and a high-temperature ($T_N \sim 960$ K) coplanar but noncollinear antiferromagnetic $L1_2$ Mn$_3$Ir phase when the Ir ratio is between 15% and 35%[20,26–32], as schematized in Fig. 1a. For bulk materials, intermetallic Mn-Ir with an Ir ratio between 35% and 40% is not stable[20]. However, for thin-film materials deposited by non-equilibrium techniques, such as sputtering and pulsed laser deposition onto oxide substrates, metastable Mn-Ir intermetallic films with Ir ratios between 35% and 40% could be successfully achieved. Once this composition boundary is obtained, the antiferromagnetic spin structure in such metastable films could be easily interconverted between the collinear and noncollinear configurations upon exposure to external stimuli, considering possible small free energy difference among the different spin structures close to the metastable composition boundary.

In order to realize the above hypothesis, we fabricated Mn-Ir thin films with a phase boundary composition of Mn$_{62.5}$Ir$_{37.5}$ utilizing magnetron sputtering onto ferroelectric 0.7PbMg$_{1/3}$Nb$_{2/3}$O$_3$-0.3PbTiO$_3$ (PMN-PT) substrates that could provide a piezoelectric strain stimulus. Accordingly, upon application of an external electric field perpendicularly to the ferroelectric PMN-PT substrate, a large nonvolatile resistance modulation of ~33% in the Mn-Ir thin film is obtained at 300 K. Systematic electrical, magnetic, and magnetotransport studies indicate that the resistance change is closely related to the spin phase change between the collinear and noncollinear structures in the Mn-Ir films. As a result, a room-temperature antiferromagnetic spin phase change memory is achieved. In addition, the resistance states of this novel memory device are highly stable over a wide temperature range, under a strong magnetic field, and for a long period. Overall, our experimental studies support our initial hypothesis of the antiferromagnetic spin phase change memory and provide a unique approach for enhancing the electrical signal outputs of antiferromagnetic spintronic devices.

## Results

### Structure and composition

The polycrystalline feature of the as-grown Mn-Ir thin film and the sharp interface between the film and the substrate can be observed via transmission electron microscopy (TEM) analysis (Supplementary Fig. 1). The composition of the Mn-Ir thin film was examined by Rutherford backscattering (RBS) technology, which is not particularly accessible, but it is accurate and widely used to determine the composition of thin films[33–36]. The experimental and fitted spectra shown in Fig. 1b revealed that the atomic concentration of Ir was approximately 37.5%. Moreover, since the PMN-PT substrate has multiple elements that could affect the RBS fitting, a Mn-Ir thin film was deposited on the Si substrate under the same conditions. This simpler RBS spectrum fitting further confirmed that the atomic concentration ratio of the Mn-Ir thin film was Mn$_{62.5}$Ir$_{37.5}$ (Supplementary Fig. 2). These results verified that composition-boundary Mn-Ir thin films were successfully fabricated[20].

### Electrical modulation

We then applied a perpendicular gate electric field $E_G$ to a Mn-Ir/PMN-PT heterostructure and examined the effect of $E_G$ on the resistance of the Mn-Ir thin film at room temperature. The measurement geometry is illustrated in Fig. 1c. It was found that the resistance of the Mn-Ir thin film is highly sensitive to $E_G$. The linear four-probe resistance as a function of $E_G$ shown in Fig. 1d exhibits a hysteretic loop and a maximum resistance variation of ~41.5%. In this measurement, we used the unipolar switching approach[13] (Fig. 1e) by applying a negative $E_G = -4$ kV·cm$^{-1}$ and a positive $E_G = +1.5$ kV·cm$^{-1}$ because ferroelectric oxides

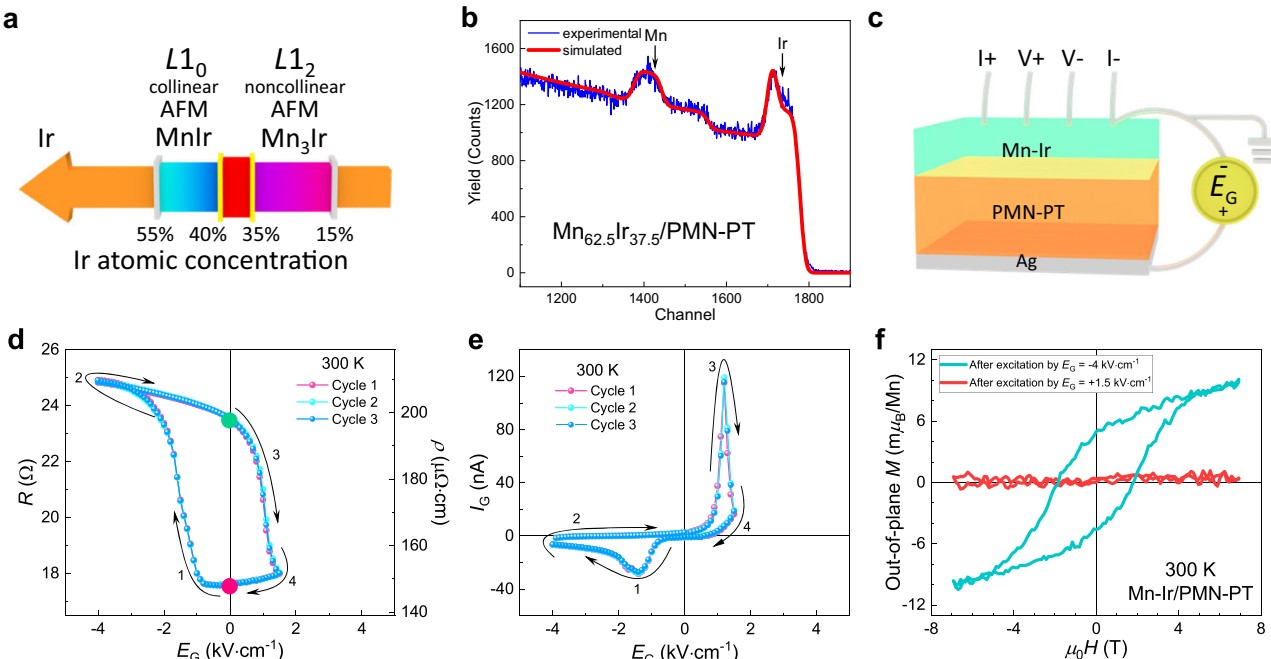

**Fig. 1 | Composition, electrical and magnetic properties of a Mn-Ir/0.7PbMg$_{1/3}$Nb$_{2/3}$O$_3$-0.3PbTiO$_3$ (PMN-PT) heterostructure. a** Illustration of the different intermetallic phases for the antiferromagnetic (AFM) Mn-Ir binary system versus the Ir atomic concentration. **b** Rutherford backscattering and fitting curves of the Mn-Ir/PMN-PT heterostructure **c** Sketch of the electric-filed ($E_G$)-gating measurement geometry for the Mn-Ir/PMN-PT heterostructure. **d** $E_G$-dependent resistance curves of the Mn-Ir thin film on the PMN-PT substrate at room temperature. **e** $E_G$-dependent gate current ($I_G$) of the PMN–PT substrate at room temperature. **f** Room-temperature out-of-plane magnetization curves of the Mn-Ir thin film after electric field pulse excitations of $E_G = -4$ and +1.5 kV·cm$^{-1}$, respectively.

tend to crack during repeated bipolar electric field cycles[37], and the unipolar switching could be effective in preventing cracking[38]. Two different states with an -5.9 Ω resistance difference at $E_G = 0$ kV·cm$^{-1}$ are observed. The nonvolatile room-temperature resistance modulation is remarkably large and is -33%.

In Fig. 1e, the gate current presents two peaks, which confirm polarization switching. On the other hand, the $E_G$-dependent resistance curve is similar to the $E_G$-dependent strain curve of PMN-PT[39–41], which indicates that the resistance modulation induced by the electric field is predominantly due to piezoelectric strain instead of electrostatic carrier injection. Specifically, if an electrostatic modulation effect exists, a positive $E_G$ will induce an enhancement of the resistance of the Mn-Ir thin film since the major carriers in the Mn-Ir thin films are determined to be holes (Supplementary Fig. 3a). In addition, the high carrier density of the Mn-Ir thin film (Supplementary Fig. 3b) corresponds to a short Thomas-Fermi electrostatic screening length of several angstroms[42]; this length is much smaller than the thickness of our Mn-Ir thin film.

To investigate the effect of the piezoelectric strain on the magnetic properties of the Mn-Ir thin film, the magnetic moments of the Mn-Ir film were measured upon different electric field pulse excitations. For example, when an $E_G$ pulse of +1.5 kV·cm$^{-1}$ is applied, the film shows negligible moment, which corresponds to a low-resistance state. In contrast, when an $E_G$ pulse of -4 kV·cm$^{-1}$ is exerted, the Mn-Ir film exhibits a small hysteresis loop with a tiny net moment of -10 m$\mu_B$/Mn (Fig. 1f). Furthermore, we deposited a $Co_{90}Fe_{10}$ (CoFe) film and a Pt capping layer on top of the Mn-Ir/PMN-PT heterostructure. Then, the exchange bias of the Pt/CoFe/Mn-Ir/PMN-PT heterostructure after applying an $E_G$ pulse excitation of -4 and +1.5 kV·cm$^{-1}$ was measured. As shown in Fig. 2, the coercivity field and the exchange bias field are -7.1 mT and -10 mT after the $E_G = +1.5$ kV·cm$^{-1}$ excitation, respectively, and then they change to 10.3 mT and 20 mT after applying an $E_G$ of −4 kV·cm$^{-1}$ pulse. The switching of the coercivity field and the exchange bias field can be repeated for several cycles. Notably, the switching of the exchange bias is less likely to be induced by the change of the CoFe layer because the magnetic properties of a Pt/CoFe/PMN-PT heterostructure without a Mn-Ir layer are insensitive to the external electric field excitations (Supplementary Fig. 4). All these results indicate that the magnetic properties of the Mn-Ir thin film can be reversibly manipulated by the piezoelectric strain.

Subsequently, we measured the Hall resistance of the Mn-Ir thin film up to 18 T at room temperature for the two nonvolatile resistance states achieved by different electric field excitations. Intriguingly, a hysteretic loop of the Hall resistance was observed and indicates the existence of the anomalous Hall effect, for the high resistance state obtained by applying an $E_G$ pulse of −4 kV·cm$^{-1}$. However, the Hall resistance curve versus magnetic field becomes a straight line for the

low-resistance state upon applying an $E_G = +1.5$ kV·cm$^{-1}$ pulse (Fig. 3a). The anomalous Hall resistance of the high-resistance state exhibits a large coercivity field and an unsaturated loop, which is similar to the previous studies of the anomalous Hall effect in $L1_2$-type noncollinear antiferromagnetic $Mn_3Ir$[31,32] with nonvanishing Berry curvature[43–45]. For noncollinear antiferromagnets exhibiting the anomalous Hall effect, a small net moment is typically measurable due to the spin canting[31,32,43–45]. While for simple collinear antiferromagnets, the anomalous Hall effect is usually absent due to the lack of spin splitting, and accordingly, the net moment is close to zero for low magnetic fields. Therefore, the magnetic data in Fig. 1f and the Hall data in Fig. 3a consistently indicate that the spin structure of the Mn-Ir thin film is reversibly interconverted by different piezoelectric strain excitations between the $L1_0$-type collinear antiferromagnetic order and the $L1_2$-type noncollinear antiferromagnetic order, as schematized in Fig. 3b, c; this is closely accompanied by a large room-temperature resistance manipulation, as shown in Fig. 1d.

## Spin phase change

To further prove the above hypothesis, X-ray absorption spectroscopy (XAS) and X-ray magnetic circular dichroism (XMCD) spectroscopy were performed on the Mn-Ir/PMN-PT heterostructure under different resistance states. Before each measurement, an in-plane magnetic field of 14 T was applied to the sample to align the possible magnetic moments. The results are shown in Fig. 4. When the Mn-Ir thin film is under the low-resistance state generated by an electric pulse of $E_G = +1.5$ kV·cm$^{-1}$, the XAS signals from left-polarized light and right-polarized light nearly overlap (Fig. 4a); thus, negligible signal is observed in the XMCD spectra (Fig. 4b), which indicates that no net magnetic moment could be detected within the detection limit. Notably, when the Mn-Ir thin film is under the high-resistance state achieved by an electric pulse of $E_G = −4$ kV·cm$^{-1}$, the XAS spectra exhibit evident differences between the two curves of left-polarized light and right-polarized light around the Mn $L_{2,3}$ edges (Fig. 4c), and both positive and negative peak signals are observed in the XMCD spectra (Fig. 4d), which indicates net magnetic moments carried by the Mn atoms. These experimental results of XAS and XMCD suggest that the collinear/noncollinear antiferromagnetic orders correspond to the high-/low-resistance states, respectively, which coincide with our magnetic and Hall characterizations in the Mn-Ir thin film and provide more evidence for the spin phase change mechanism by electric fields.

Additionally, the TEM images of the Mn-Ir/PMN-PT heterostructure under both the high- and low-resistance states indicate that the lattice of the Mn-Ir thin film under the high-resistance state is closer to the cubic $L1_2$ noncollinear antiferromagnetic phase of the Mn-Ir system. In contrast, the lattice of the Mn-Ir thin film under the low-resistance state is closer to the tetragonal $L1_0$ collinear

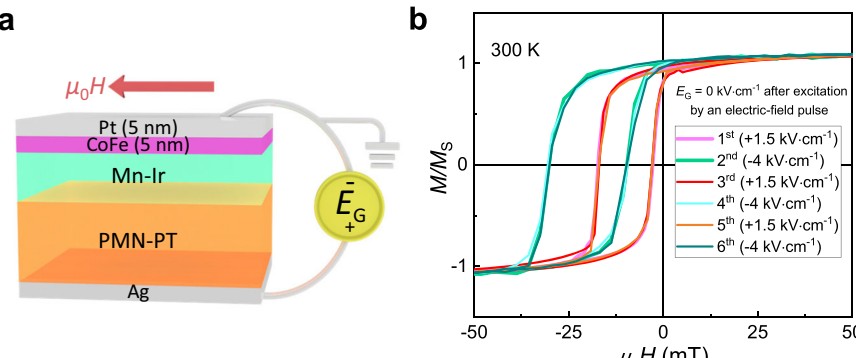

**Fig. 2 | Electric-field-controlled exchange bias effect in a Pt/$Co_{90}Fe_{10}$ (CoFe)/ Mn-Ir/PMN-PT heterostructure. a** Schematics of the Pt/CoFe/Mn-Ir/PMN-PT heterostructure, the exchange bias measurement geometry and the gate electric field switching geometry. **b** Room-temperature exchange bias of the heterostructure shown in (**a**) under the high-resistance state (HRS)/low-resistance state (LRS) excited by sequential positive/negative electric field pulses.

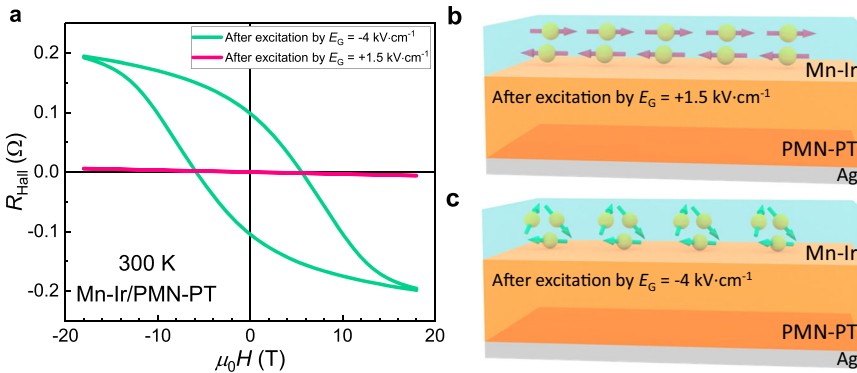

**Fig. 3 | Manipulation of the Hall resistance at room temperature and possible spin phase change mechanism by electric fields for the Mn-Ir/PMN-PT film.** **a** Room-temperature Hall effect of the Mn-Ir thin film under the HRS/LRS attained by pulses of $E_G = -4$ and $+1.5$ kV·cm⁻¹, respectively. **b** Diagrammatic sketch of the collinear antiferromagnetic spin structure of the $L1_0$-type phase in the Mn-Ir thin

film under the LRS triggered by an electric pulse of $E_G = +1.5$ kV·cm⁻¹. **c** Diagrammatic sketch of the noncollinear antiferromagnetic spin structure of the $L1_2$-type phase in the Mn-Ir thin film under the HRS triggered by an electric pulse of $E_G = -4$ kV·cm⁻¹.

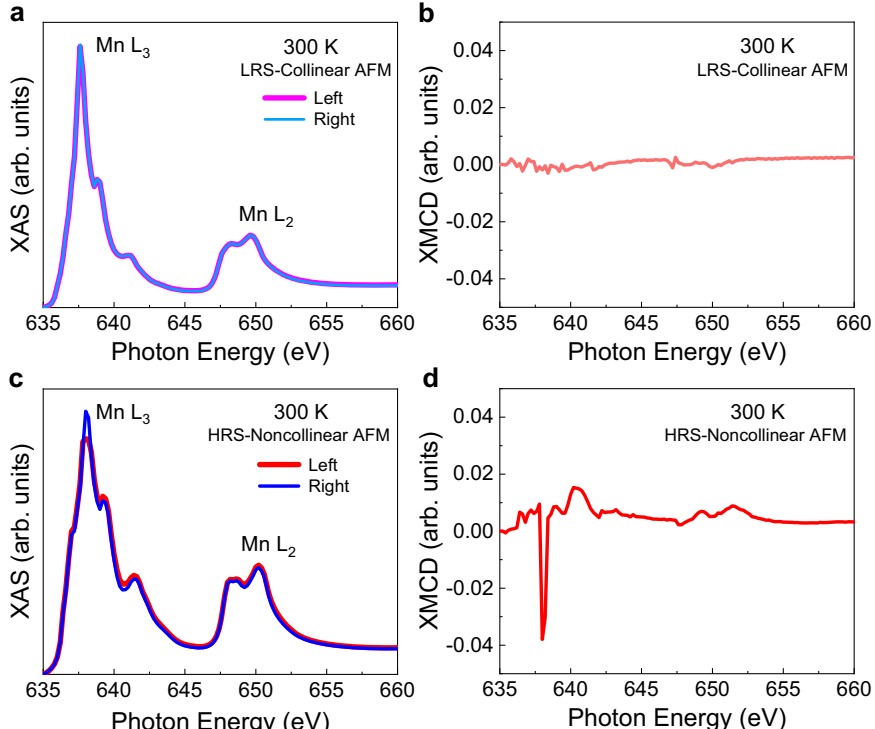

**Fig. 4 | X-ray absorption spectra (XAS) and X-ray magnetic circular dichroism (XMCD) of the Mn-Ir/PMN-PT heterostructure under different resistance states. a, b** XAS (**a**) and XMCD (**b**) of the Mn-Ir/PMN-PT heterostructure under the LRS excited by an electric pulse of $E_G = +1.5$ kV·cm⁻¹. **c, d** XAS (**c**) and XMCD (**d**) of

the Mn-Ir/PMN-PT heterostructure under the HRS realized by an electric pulse of $E_G = -4$ kV·cm⁻¹. The data were collected around manganese $L_{2,3}$ edges at room temperature (300 K).

antiferromagnetic phase (Fig. 5). Moreover, the variations in the lattice parameters of the Mn-Ir thin film under the high- and low-resistance states are in agreement with the compressive and tensile strains induced by the PMN-PT substrate after applying gate electric fields of $E_G = -4$ kV·cm⁻¹ and $+1.5$ kV·cm⁻¹, respectively.

Furthermore, $L1_2$ noncollinear antiferromagnetic Mn₃Ir (atomic ratio: Mn/Ir ∼ 75/25) thin films and $L1_0$ collinear antiferromagnetic MnIr (atomic ratio: Mn/Ir ∼ 50/50) thin films were fabricated as reference thin films. The noncollinear $L1_2$ Mn₃Ir thin film exhibits an anomalous Hall effect and an uncompensated net magnetization, which is similar to that of the Mn-Ir thin film under the high-resistance state. Moreover, no anomalous Hall effect and no net

magnetization are detected in the collinear $L1_0$ MnIr thin film, which is identical to that of the Mn-Ir thin film in the low-resistance state. The resistance values of the Mn₃Ir and MnIr thin films detected at room temperature with similar electrical measurement geometries are close to the high- and low-resistance states of the Mn-Ir thin film, respectively (Supplementary Fig. 5). These results support that the Mn-Ir thin film clearly changes between its tetragonal and cubic phases due to the piezoelectric strain of the PMN-PT substrate generated via the gate electric field.

To obtain a reasonable physical picture of the spin phase change in the Mn-Ir thin film induced by piezoelectric strain, theoretical simulations on the free energies of the different spin configurations

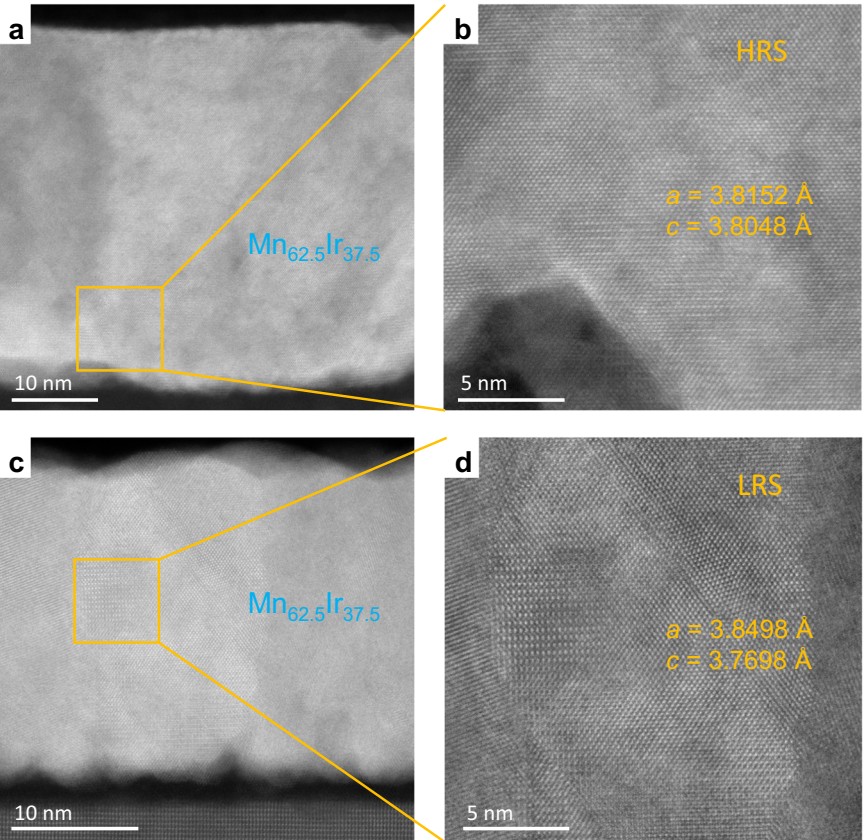

**Fig. 5 | Transmission electron microscopy (TEM) images of the Mn-Ir/PMN-PT heterostructure under different resistance states. a, b** Cross-sectional TEM image (**a**) and magnified image (**b**) of the Mn-Ir/PMN-PT heterostructure under the HRS generated via an electric pulse of $E_G = -4\ kV\cdot cm^{-1}$. **c, d** Cross-sectional TEM image (**c**) and magnified (**d**) of the Mn-Ir/PMN-PT heterostructure under the LRS excited by an electric pulse of $E_G = +1.5\ kV\cdot cm^{-1}$.

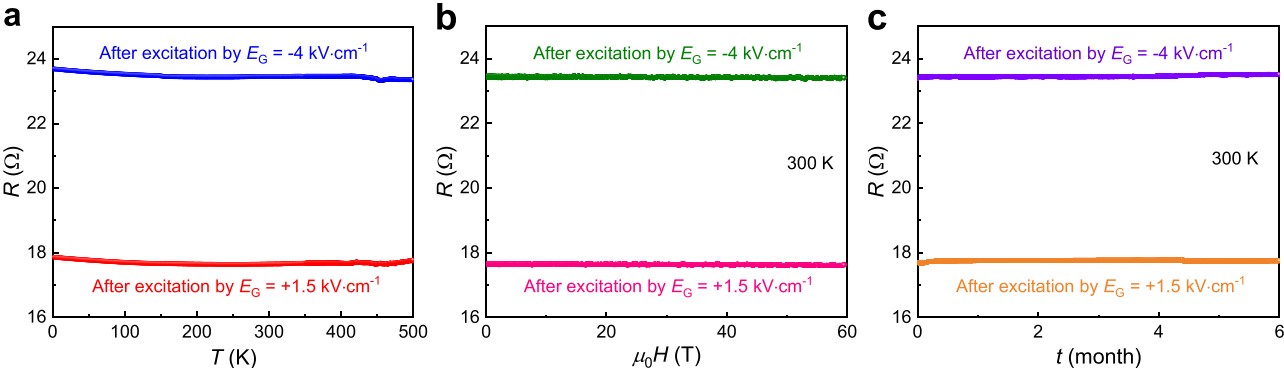

**Fig. 6 | Temperature, magnetic field, and time stability of the Mn-Ir based antiferromagnetic spin phase change memory. a** Temperature dependence of the HRS/LRS from 2 to 500 K. **b** Magnetic field dependence of the HRS/LRS under strong pulsed magnetic fields up to 60 T at room temperature. **c** Time stability of the HRS/LRS of the memory during a 6-month exposure to air.

were performed. As elaborated in Supplementary Note 1, we theoretically identified a reasonable atomic configuration (Supplementary Fig. 6a) corresponding to the complex microstructure of the composition-boundary Mn-Ir thin film. For this crystal structure, the difference in the free energies of the collinear (Supplementary Fig. 6b) and noncollinear spin (Supplementary Fig. 6c) configurations is rather small, at only ~42 meV/unit cell. Thus, for a Mn-Ir film with an atomic composition between the stable collinear $L1_0$ and noncollinear $L1_2$ phases, subtle external energetic stimuli could easily lead to the interconversion between the collinear and noncollinear antiferromagnetic spin phases. These theoretical studies validate the physical principle of the new antiferromagnetic spin phase change memory.

On the other hand, the band structures and density of states (DOSs) for both collinear and noncollinear phases were also calculated (Supplementary Fig. 7). The results demonstrate that around the Fermi level, the electronic DOSs in the noncollinear phase are significantly lower than those in the collinear phase. This indicates that more conducting channels can be observed in the collinear phase. These findings are consistent with the experimental results, qualitatively illustrating the low-resistance characteristics of the collinear phase and the high-resistance characteristics of the noncollinear phase.

## Memory Stability

To examine the stability of this spin phase change memory device, we measured the changes in the resistance states relative to temperature, magnetic field, and time. As shown in Fig. 6a, both the high- and low-resistance states subtly vary over a large temperature range of ~500 K. This excellent temperature stability of the resistive states originates from the transport feature that is likely impacted by multiple mechanisms in the Mn-Ir system, which exhibits a rather weak temperature dependence of the resistivity, as shown in Supplementary Fig. 3c. More importantly, the difference between the high- and low-resistance states can be differentiated over the entire temperature range, which indicates that this memory device is capable of working over a large temperature range even above 200 °C. Figure 6b shows plots of the high- and low-resistance states versus the strong pulsed magnetic field. Remarkably, the two resistance states are rather robust over the perturbations of the pulsed magnetic field up to 60 T. This superior feature is a consequence of the combination of the rather high antiferromagnetic ordering temperature of Mn-Ir and the low carrier mobility (Supplementary Fig. 3d) that suppresses orbital scattering due to the Lorentz force. These results demonstrate that the Mn-Ir-based memory device can resist a high magnetic field up to 60 T. In addition, even throughout six months, no remarkable changes were observed for the high- and low-resistance states, which confirms the excellent time stability of the device (Fig. 6c). Overall, this antiferromagnetic spin phase change memory exhibits ultrahigh reliability upon temperature, magnetic field, and time variations.

## Discussion

In summary, we have successfully fabricated composition-boundary Mn-Ir thin films onto functional ferroelectric PMN-PT substrates. At the composition boundary, the spin structure of the Mn-Ir films was highly sensitive to external stimuli. Upon piezoelectric excitations triggered by different electric fields, the antiferromagnetic spin structure is reversibly changed between a noncollinear and a collinear antiferromagnetic phase, leading to a large nonvolatile resistance variation of ~33%. Moreover, the resistance states were superiorly stable over a large temperature range of ~500 K, an ultrahigh magnetic field of 60 T or a long half-year time. Overall, our study has achieved a novel antiferromagnetic spin phase change memory with large electrical outputs and excellent reliabilities, mimicking the conventional phase change memory from the quantum spin degree of freedom.

In addition, the write cycle endurance in conventional phase change memories is typically limited by atomic migration characteristics[46]. However, according to the theoretical simulations, no atomic rearrangement occurs during the phase change in the Mn-Ir spin phase change memory, which indicates an unlimited write cycle endurance compared to that of conventional phase change memories. Moreover, the Mn-Ir system controlled by piezoelectric strain has potential applications in all-antiferromagnetic tunnel junctions as electrodes, which could introduce additional memory states and realize multiple-state storage[47–49], thus providing a novel method for high-density storage and artificial synapses.

## Methods

### Growth

Mn-Ir thin films were grown onto (001)-oriented $0.7PbMg_{1/3}Nb_{2/3}O_3$–$0.3PbTiO_3$ (PMN-PT) substrates using a d.c. sputtering technique with a $Mn_{68}Ir_{32}$ polycrystalline target. The base pressure of the sputtering system was $7.5 \times 10^{-9}$ Torr. The deposition procedure was implemented at 500 °C with a sputtering power of 60 W and an Ar gas pressure of 3 mTorr. For the Pt/$Co_{90}Fe_{10}$/Mn-Ir/PMN-PT and Pt/$Co_{90}Fe_{10}$/PMN-PT heterostructures, both the $Co_{90}Fe_{10}$ films and the Pt films were deposited at room temperature in a d.c. the sputtering system using a $Co_{90}Fe_{10}$ polycrystalline target and a Pt target, respectively, with an Ar pressure of 3 mTorr. The growth rate of the $Co_{90}Fe_{10}$ thin film was 0.29 Å·s$^{-1}$ with a sputtering power of 90 W, and the growth rate of the Pt thin film was 0.28 Å·s$^{-1}$ with a sputtering power of 30 W.

### Rutherford backscattering

Rutherford backscattering spectrometry of the Mn-Ir thin films on the PMN-PT substrate and the Si substrate was performed at the Centre for Ion Beam Applications at the National University of Singapore. The energy of He$^+$ is 1.5 MeV, and the two detectors are positioned at 30 degrees and 90 degrees.

### Transmission electron microscopy

The cross-section samples of the thin-film heterostructures were prepared by the focused ion beam technique. The transmission electron microscopy images were collected by an ARM 200CF (JEOL, Tokyo, Japan) operated at 200 kV. HAADF images were acquired at an acceptance angle of 90–370 mrad.

### Electrical measurements

A standard four-probe measurement geometry was used for the longitudinal resistance measurements, and the contacts were made by a wire bonding with Al wires. The electrical measurements were carried out in a Quantum Design VersaLab system utilizing the Delta measurement model supported by a Keithley 6221 current source and a Keithley 2182 A nanovoltmeter. Perpendicular gate electric fields were applied through another Keithley 2400 source meter. The standard Hall measurement geometry was used for the Hall measurements.

### Magnetic measurements

Magnetic measurements were carried out in a Quantum Design superconducting quantum interference device.

### X-ray absorption spectroscopy

X-ray absorption and X-ray magnetic circular dichroism spectroscopy of the Mn-Ir thin film under different resistance states were collected at beamline 07U of the Shanghai Synchrotron Radiation Facility. The grazing incidence angle was 45°, and an in-plane magnetic field of 0.5 T was applied. The characteristic manganese $L_{2,3}$ edge peaks were specifically measured.

### High magnetic field transport measurements

Linear four-probe electrical contacts were made by silver paint and Au wires. The measurements were conducted at the Wuhan National High Magnetic Field Center. A 3 mA *a.c.* current with a frequency of 89 kHz was injected by a National Instruments PXI-5402 waveform generator for resistance measurements. The resulting voltage was measured by a National Instruments PXIe-5105 oscilloscope at a frequency of 4 MHz. To increase the accuracy of the high-frequency electrical resistance measurements, the middle two-probe voltage was amplified 100 times using a Stanford Research SR560 preamplifier.

## Data availability

All the data that support the plots within this paper are available from the corresponding authors upon request.

## Code availability

All the simulation codes in this study are available from the corresponding authors upon request.

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

## Acknowledgements

Z.L. acknowledges the financial support of the National Key R&D Program of China (Grant No. 2022YFA1602700). Z.L. acknowledges the financial support of the National Key R&D Program of China (Grant No. 2022YFB3506000). Z.L. & C.J. acknowledge financial support from the National Natural Science Foundation of China (Grant No. 52121001). Z.L. acknowledges financial support from the National Natural Science Foundation of China (Grant No. 52271235). Z.L. acknowledges financial support by Beijing Natural Science Foundation (Grant No. JQ23005). P.T. was supported by the National Natural Science Foundation of China (Grants No. 12234011 and No. 12374053). Q.Z. acknowledges the National Natural Science Foundation of China (Grants No. 52322212 and No. 52072400). P.Q. acknowledges the financial support of the China National Postdoctoral Program for Innovative Talents (Grant No. BX20230451).

## Author contributions

H.Y. and H.M. contributed equally to this work. H.Y. performed sample growth and electrical and magnetic measurements, with assistance from P.Q., X.Z., X.W., H.C., Z.M., L.L., G.Z., and Z.D. J.W. and Z.Zhu performed the High magnetic field transport measurements. H.L. and A.A.B performed the RBS measurements. B.F. and Z.Zeng performed the patterned device fabrication. Q.Z. performed the TEM measurements. H.M. and P.T. performed the theoretical calculations. H.Y. wrote the manuscript, along with P.Q., X.Z., X.W., H.C., Z.M., L.L., and Z.L. All authors discussed the results and commented on the manuscript. Z.L.,C.J. and P.Q. conceived and led the project.

## Competing interests

The authors declare no competing interests.
