## [Peer Review File · Nature Communications]

Reviewers' Comments:

Reviewer #1:

Remarks to the Author:

I have reviewed the journal article that uses a piezoelectric sample to transform the collinear antiferromagnetic phase into a noncollinear antiferromagnetic phase and vice versa. The authors have provided several studies supporting this work including measurements of resistance change, shifts in exchange bias (with FeCo), XMCD data, TEM data, and some thermal studies. The work also indicates that a phase change occurs between a cubic state and a tetragonal state during the application of the electric field, and vice versa. While the work provides several complimentary studies this reviewer still has some questions that must be answered in the manuscript prior to this work being published. Below are the questions from this reviewer.

1. Authors need to provide the orientation of the PMN-PT sample (e.g. 001 or 011 cut) to the community, i.e. these two cuts provide dramatically different response. Authors also need to describe when they performed the initial poling of the sample, i.e. before or after depositions.
2. This reviewer is aware of strains or stresses used to induce phase changes in a wide variety of materials including antiferromagnetic to ferromagnetic materials, e.g. shifts the transformation temperatures. However, typically once the stresses or strain are removed in most cases this phase remains and the reversal does not occur. Authors need to provide a compelling story why their system is able to be transitioned back and forth between two stable states with stresses or strains. This explanation needs to include the fact that the authors report each of these phases is stable over a very broad temperature range (Figure 6). While the results from the authors are compelling data I have a hard time resolving how you can have both thermal stability of these phases as well as the ability to transform between the two phases at one temperature with stresses.
3. As a follow up to the above, one can rotate magnetic spins in magnetostrictive materials and have this remain stable at different strains and temperatures but note this does not require a phase transformation.
4. Authors also need to provide a brief description of why strains/stresses can cause this phase transformation but other fields, e.g. magnetic to thermal do not.

In conclusion this reviewer finds the work extremely interesting but is concerned about the physics of explaining what is happening.

Reviewer #2:

Remarks to the Author:

In this work, the authors have investigated an antiferromagnetic spintronic device based on an Ir-Mn alloy film on a piezoelectric substrate, and found that the giant changes of the resistance and anomalous Hall resistance can be induced by an electric field. This is attributed to the transition between the collinear and noncollinear antiferromagnetic phases driven by the piezoelectric strain. The ON/OFF ratio induced by the phase transition is significantly larger than that in the reported devices based on a single antiferromagnetic film. They have also shown the device have a desirable stability. This is an exciting discovery, which opens a new direction of antiferromagnetic spintronics with low energy dissipation and high ON/OFF ratio by exploiting the antiferromagnets close to the magnetic phase boundary. I therefore recommend the publication of this work on Nature Communications.

I have some minor comments/suggestions, as listed below:

1. The authors have calculated the energy of the collinear and noncollinear Ir-Mn system using the

experimental lattice parameters. I suggest they also calculate the energy using the fully optimized lattice parameters. The calculations using different parameters will offer direct evidence that the energies of these phases can be strongly remodulated by the strain.

2. I suggest the authors show the band structures and density of states of these two phases in the supplemental material, which can be used to qualitatively explain the different longitudinal transport properties of these two phases.

3. Using such a system as the electrodes in an antiferromagnetic tunnel junction would be promising. The conventional MTJs/AFMTJs have only two resistance states of P and AP configuration by the spin matching mechanism (arXiv:2312.13507). The magnetic phase transition of the electrodes could introduce additional memory states, and open an opportunity to realize a high storage density based on AFMTJs. I suggest the author discuss this.

Reviewer #3:

Remarks to the Author:

The work of Yan et al. describes the results of an experimental study backed by theoretical work that for a narrow compositional range (35-40% Ir) of Mn-Ir thin films that show switchable colinear and non colinear antiferromagnetic ordering in response to a change in strain state. The ability to change strain state comes from growing the film on a PMN-PT piezoelectric substrate where an applied perpendicular electric field changes induces a small change in dimensions. There are many positive features of this work but also queries which will need to be carefully addressed. The written quality is currently not at an acceptable level with some inappropriate wording and phraseology choices. The work should be thoroughly proof-read prior to any resubmission.

Major points:

1. The introduction should mention the current (important) use of antiferromagnetic thin films for exchange bias in, for example, spin valves not as auxiliary materials.
2. The introduction notes that "For bulk materials, the intermetallic Mn-Ir with a Ir ratio between 35% and 40% is not stable". There are no references and it is quite unclear what "not stable" means – presumably it does not combust spontaneously! State if the ratio quoted is atomic percent.
3. There is no information about the PMN-PT substrates. It is generally recognized that these substrates can be quite difficult to work with given their complex crystallographic and compositional structure. I would expect to see some basic description/characterisation of the substrates, possibly in the supplementary information. One explanation for the very large fields required is that there is a significant distribution of substrate properties.
4. There should be references to the use of Rutherford Backscattering (RBS) in the analyses of thin films. This is not a particularly common technique and it is important that readers are able to understand its uses and limitations.
5. The authors present some very nice TEM results but without full experimental detail – how were the samples prepared, I presume a cut out lamella?
6. There are no XRD data. This seems an obvious and significant omission since typically XRD, and possibly XRR, are the first structural measurements done to characterize thin films.
7. The film thickness is not given. This is important as the strain-driven changes to magnetic ordering can occur only within a finite distance of the interface. Do the authors have any estimate for the distance over which the strain-mediate effects occur?
8. When applying +1.5 kV/cm to the sample with CoFe the resistance state is low, the spins are colinear, and the exchange bias is 10 mT. On applying -4.0 kV/cm the resistance state is high, the spins are non-colinear, and the exchange bias increases to 20 mT. Naïvely, I would have guessed that the exchange bias is more significant for the case of colinear AF, where there is no field-

induced spin canting. Do the authors have any intuition as to why the non-collinear AF exhibits a larger exchange bias effect?

9. The AHE measurements show the coercivity of the spin-canted L12 noncolinear phase to be substantially higher than the magnetometry data of Fig. 1f (>5 T compared to ~ 2 T). What is the reason for this difference?

10. Figure 4 needs to have a legend.

11. The TEM images are a particularly nice demonstration of the structural phase change. Do they show any variations with distance from the PMN-PT substrate, either in tetragonality or in the magnitude of the strain? The two regions sampled to calculate the lattice parameters are at different distances from the substrate; it is important to address the degree to which this may be responsible for the observed differences.

12. What is the dark contrast region between the Mn-Ir and the highly crystalline PMN-PT substrate seen in the TEM images?

13. The dimensions used of the electrical measurements should be given – the substrate size, probe spacing etc. Were wires directly bonded to the Mn-Ir film? Was the same setup used for both the Mn-Ir and the Mn-Ir/CoFe samples?

14. The data in fig.6a are described as "...originates from the dirty transport feature...". This is incomprehensible, I have no idea what "dirty transport feature" means.

15. The data in Fig. 6c shows that the resistance states are stable over a long period of time (6 months). Was the resistance state sampled at several times within this period, or just at the start and the end? The line on the figure appears to have some noise on it, suggesting a large number of measurements were made.

16. The calculations in the supplementary information show that the spin reorientation can occur without atomic rearrangement (e.g. due to dislocation formation with strain), a particularly nice result given that the write cycle endurance in conventional phase change memories is typically limited by atomic migration characteristics. The main text could go further to highlight this point, I think.

17. The supplementary information consists of a series of figures with no commentary or experimental explanations. This is unsatisfactory and must be addressed in any resubmission.

Response to the Reviewers' Comments

Reviewer #1

I have reviewed the journal article that uses a piezoelectric sample to transform the collinear antiferromagnetic phase into a noncollinear antiferromagnetic phase and vice versa. The authors have provided several studies supporting this work including measurements of resistance change, shifts in exchange bias (with FeCo), XMCD data, TEM data, and some thermal studies. The work also indicates that a phase change occurs between a cubic state and a tetragonal state during the application of the electric field, and vice versa. While the work provides several complimentary studies this reviewer still has some questions that must be answered in the manuscript prior to this work being published. Below are the questions from this reviewer.

Response: We would like to express our sincere gratefulness to the Reviewer for her/his comprehensive summarization and valuable comments on our manuscript.

1. Authors need to provide the orientation of the PMN-PT sample (e.g. 001 or 011 cut) to the community, i.e. these two cuts provide dramatically different response. Authors also need to describe when they performed the initial poling of the sample, i.e. before or after depositions.

Response: The PMN-PT substrates used in this work are (001)-oriented. And the PMN-PT substrates did not undergo any polarization before the thin-film deposition. We have added the information about the PMN-PT substrates in the supplementary information as "Supplementary Note 2. Ferroelectric substrates".

2. This reviewer is aware of strains or stresses used to induce phase changes in a wide variety of materials including antiferromagnetic to ferromagnetic materials, e.g. shifts the transformation temperatures. However, typically once the stresses or strain are removed in most cases this phase remains and the reversal does not occur. Authors need to provide a compelling story why their system is able to be transitioned back and forth between two stable states with stresses or strains. This explanation needs to include the fact that the authors report each of these phases is stable over a very broad temperature range (Figure 6). While the results from the authors are compelling data I have a hard time resolving how you can have both thermal stability of these phases as well as the ability to transform between the two phases at one temperature with stresses.

Response: The phase and spin order of the Mn-Ir thin films is closely related to its crystal structure and lattice constants. Specifically, the collinear antiferromagnetic $L1_0$ MnIr phase has a tetragonal structure with $a = b = 3.855 \text{ \AA}$ and $c = 3.644 \text{ \AA}$. The noncollinear antiferromagnetic $L1_2$ Mn₃Ir phase has a cubic structure with $a = b = c = 3.778 \text{ \AA}$.

Moreover, the polarization of the PMN-PT substrates can be reversibly switched by the positive and negative electric fields, as can the piezoelectric strain. This strain in PMN-PT is elastic and reversible, which not only can reversibly manipulate the magnetic structure and electrical transport properties [C. Thiele et al. *Phys. Rev. B*, **75**, 054408 (2007); S. Zhang et al. *Phys. Rev. Lett.*, **108**, 137203 (2012); S. Z. Wu et al. *Sci. Rep.*, **5**, 8905 (2015); J.-M. Hu et al. *Adv. Mater.*, **28**, 15 (2016); B. Peng et al. *ACS Nano*, **11**, 4337 (2017); W. Hu et al. *Nat. Nanotech.*, **14**, 668 (2019); X. Chen et al. *Nat. Mater.*, **18**, 931 (2019)] in the thin films but also can reversibly tune the light and

ultrasound emissions [Y. Zhang et al. *Adv. Mater.*, **24**, 1729 (2012); Y. Zhang et al. *Adv. Mater.*, **30**, 1707007 (2018); J. Yang et al. *J. Semicond.*, **41**, 011901 (2020)].

Thus, in this work, when the electric-field-induced piezoelectric strain exerts on the Mn-Ir thin films deposited onto the PMN-PT substrates, the lattice constants of Mn-Ir can be reversibly manipulated (Fig. 5), which leads to the reversible interconversion between the collinear phase and the noncollinear phase. When the gate electric field is removed, this piezoelectric strain will be maintained and not disappear, that is, this modulation is non-volatile. And when applying a gate electric field with opposite polarity, there will be a reversal.

On the other hand, other stimulations like magnetic fields and heat cannot cause the phase change in the Mn-Ir thin films. Because the Néel temperatures of both the $L1_0$ -collinear phase (~1145 K) and the $L1_2$ -noncollinear phase (~960 K) of the Mn-Ir alloys are extremely high, which means high stability upon temperature and magnetic fields.

Overall, the phase transition in the Mn-Ir thin films is closely related to its crystal structure and highly sensitive to the change of lattice constants induced by the piezoelectric strain but robust to heat and magnetic fields.

3. As a follow up to the above, one can rotate magnetic spins in magnetostrictive materials and have this remain stable at different strains and temperatures but note this does not require a phase transformation.

Response: We fully agree the Reviewer that the magnetic spins can be controlled via magnetostriction. It is also an effective way of strain modulation and it has completely different mechanism from this work.

4. Authors also need to provide a brief description of why strains/stresses can cause this phase transformation but other fields, e.g. magnetic to thermal do not.

Response: Thanks for the nice point. As we explained above, this phase transformation is greatly sensitive to the change of lattice constants, which could be effectively and reversibly manipulated by the electric-field-induced piezoelectric strain. The strong resistance to magnetic-field and thermal excitations in Mn-Ir system could be attributed to its strong antiferromagnetic exchange coupling, high Néel temperature and the transport feature that is likely impacted by multiple mechanisms. Besides, the brief descriptions about the points have been within the manuscript.

“This excellent temperature stability of the resistive states originates from the transport feature that is likely impacted by multiple mechanisms in the Mn-Ir system, which exhibits a rather weak temperature dependence of the resistivity”

“Remarkably, the two resistance states are rather robust over the perturbations of the pulsed magnetic field up to 60 T. This superior feature is a consequence of the combination of the rather high antiferromagnetic ordering temperature of Mn-Ir and the low carrier mobility that suppresses the orbital scattering due to the Lorentz force.”

In conclusion this reviewer finds the work extremely interesting but is concerned about the physics of explaining what is happening.

Response: We sincerely appreciate the Reviewer again for her/his encouragement and valuable opinions on our work. Motivated by these, we have carefully refined our work in physics. Hopefully, we believe that our explanation can dispel the Reviewer's concerns.

Reviewer #2

In this work, the authors have investigated an antiferromagnetic spintronic device based on an Ir-Mn alloy film on a piezoelectric substrate, and found that the giant changes of the resistance and anomalous Hall resistance can be induced by an electric field. This is attributed to the transition between the collinear and noncollinear antiferromagnetic phases driven by the piezoelectric strain. The ON/OFF ratio induced by the phase transition is significantly larger than that in the reported devices based on a single antiferromagnetic film. They have also shown the device have a desirable stability. This is an exciting discovery, which opens a new direction of antiferromagnetic spintronics with low energy dissipation and high ON/OFF ratio by exploiting the antiferromagnets close to the magnetic phase boundary. I therefore recommend the publication of this work on Nature Communications.

Response: We sincerely thank the Reviewer for the comprehensive summarization of our work. Particularly, we would like to express our heartfelt gratitude for the high praise of the “exciting discovery” and significant suggestions. Motivated by these, we have improved our work a lot, especially the theoretical part.

I have some minor comments/suggestions, as listed below:

1. The authors have calculated the energy of the collinear and noncollinear Ir-Mn system using the experimental lattice parameters. I suggest they also calculate the energy using the fully optimized lattice parameters. The calculations using different parameters will offer direct evidence that the energies of these phases can be strongly remodulated by the strain.

Response: We thank the reviewer for his/her nice suggestions, which help us a lot to improve the quality of this work. Following the reviewer’s suggestions, we optimized the lattice structure of Mn_3Ir by using the GGA-PBE exchange-correlation functional in the framework of DFT calculations. It is found that the optimized lattice constant ($a_{\text{opt}} = 3.69\text{\AA}$) is much smaller than the experimental value ($a_{\text{exp}} = 3.84\text{\AA}$). Then, we used the optimized lattice to construct the supercell for Ir-Mn system, in which some Mn atoms were substituted by the Ir atoms. The optimized lattice parameters for the supercell are $a = 3.688\text{\AA}$, $b = 18.44\text{\AA}$, and $c = 7.376\text{\AA}$, in contrast to the experimental lattice parameters of $a = 3.84\text{\AA}$, $b = 19.2\text{\AA}$, and $c = 7.54\text{\AA}$. Then we compared the formation energy for these structures with collinear and noncollinear magnetic configurations. The calculated results are shown in Fig. R1. Within the experimental lattice parameters, the noncollinear phase exhibits lower energy, while within a fully optimized lattice parameters, the collinear phase exhibits lower energy. As the reviewer mentioned, such results clearly demonstrate that the formation energy of Mn-Ir system can be strongly remodulated by the strain, leading to different spin configurations.

Fig. R1 Theoretical modeling of the possible atomic and spin configurations. **a**, A relaxed crystal structure pertaining to a composition-boundary Mn-Ir film. **b**, The collinear antiferromagnetic phase of the experimental lattice parameters and the fully optimized lattice parameters in (a) with the formation energy ($E_{\text{exp-f}}$, $E_{\text{opt-f}}$). **c**, The noncollinear antiferromagnetic phase of the experimental lattice parameters and the fully optimized lattice parameters in (a) with the formation energy ($E_{\text{exp-f}}$, $E_{\text{opt-f}}$).

2. I suggest the authors show the band structures and density of states of these two phases in the supplemental material, which can be used to qualitatively explain the different longitudinal transport properties of these two phases.

Response: This is a greatly helpful suggestion. Following the reviewer's nice suggestions, we calculated the band structures and density of states (DOSs) for both collinear and noncollinear phases within the lattice configuration shown in in Supplementary Figure 6a. The results demonstrate that around the Fermi level, the electronic DOSs in the noncollinear phase is significantly lower than that in the collinear phase. This indicates that more conducting channels can be observed in the collinear phase. These findings are consistent with experimental results, qualitatively illustrating the low-resistance characteristics of the collinear phase and the high-resistance characteristics of the noncollinear phase.

Fig. R2 the band structures and density of states for both collinear and noncollinear phases. **a**, The band structure for the collinear phase. **b**, The band structure for the noncollinear phase. **c**, The comparison of the density of states between collinear and noncollinear phases.

3. Using such a system as the electrodes in an antiferromagnetic tunnel junction would be promising. The conventional MTJs/AFMTJs have only two resistance states of P and AP configuration by the spin matching mechanism (arXiv:2312.13507). The magnetic phase transition of the electrodes could introduce additional memory states, and open an opportunity to realize a high storage density based on AFMTJs. I suggest the author discuss this.

Response: We appreciate the Reviewer for the constructive opinion. We have added a discussion about the potential of applications in antiferromagnetic tunnel junctions.

“On the other hand, the Mn-Ir system controlled by piezoelectric train is promising to be utilized in all-antiferromagnetic tunnel junctions as the electrodes, which could introduce additional memory states and realize multiple-state storage⁴⁷⁻⁴⁹, thus give a brand-new way for high-density storage and artificial synapse.”

Reviewer #3

The work of Yan et al. describes the results of an experimental study backed by theoretical work that for a narrow compositional range (35-40% Ir) of Mn-Ir thin films that show switchable colinear and non colinear antiferromagnetic ordering in response to a change in strain state. The ability to change strain state comes from growing the film on a PMN-PT piezoelectric substrate where an applied perpendicular electric field changes induces a small change in dimensions.

There are many positive features of this work but also queries which will need to be carefully addressed. The written quality is currently not at an acceptable level with some inappropriate wording and phraseology choices. The work should be thoroughly proof-read prior to any resubmission.

Response: We are sincerely grateful to the Reviewer for her/his positive comments on our work. More importantly, the thoughtful comments and suggestions provided by the Reviewer are extremely valuable to improve our work.

In terms of the English writing, we have consulted a professional language editing company associated with the Nature Publishing Group to improve the main text (Fig. R3).

Fig. R3 Printsreen of the English editing certificate.

Major points:

1. The introduction should mention the current (important) use of antiferromagnetic thin films for exchange bias in, for example, spin valves not as auxiliary materials.

Response: We apologize for the misunderstanding caused by our inaccurate words. We do not think that antiferromagnetic materials are not important, but compared to ferromagnetic materials, they are not currently used as a core functional layer to storage the information but a pinning layer in devices such as spin valves. We have revised the relevant sentences in the introduction.

“For more than 70 years since its discovery by M. Louis Néel¹ in 1948, antiferromagnets had remained underutilized as the key functional materials in spintronic devices. Instead, they had been primarily employed as pinning layers in devices such as spin valves because of their inherent net zero magnetic moments and ability to induce exchange bias with ferromagnetic materials.”

2. The introduction notes that “For bulk materials, the intermetallic Mn-Ir with a Ir ratio between 35% and 40% is not stable”. There are no references and it is quite unclear what “not stable” means – presumably it does not combust spontaneously! State if the ratio quoted is atomic percent.

Response: Mn-Ir is an intermetallic compound, which means that unlike solid solution, it has a defined stoichiometry and crystal structure with designated sites assigned for the atoms of each constituent element [H. Okamoto et al. *J. Phase Equilib.*, **17**, 60 (1996)]. And we have added the reference into the manuscript at proper position. Thus, when the Ir ratio is between 35% and 40%, the bulk Mn-Ir is unstable and will form enriched MnIr phase in some regions and Mn atom enrichment in some regions, or enriched Mn₃Ir phase in some regions and Ir atom enrichment in some regions, but there will be no uniform Mn-Ir system with large area. So, it is called “not stable”. However, for thin films, it is feasible to obtain MnIr with the Ir ratio between 35% and 40% by non-equilibrium techniques such as sputtering and pulsed laser deposition onto oxide substrates, and we did successfully achieve it in the experiment.

In addition, the ratio quoted is atomic percent, which we had interpreted as “Ir atomic concentration” in the Fig. 1a.

3. There is no information about the PMN-PT substrates. It is generally recognized that these substrates can be quite difficult to work with given their complex crystallographic and compositional structure. I would expect to see some basic description/characterisation of the substrates, possibly in the supplementary information. One explanation for the very large fields required is that there is a significant distribution of substrate properties.

Response: We fully agree the Reviewer that the PMN-PT substrates are important in this work. PMN-PT substrates are a kind of ferroelectrics with excellent piezoelectric performance, through which it not only can reversibly manipulate the magnetic structure and electrical transport properties [C. Thiele et al. *Phys. Rev. B*, **75**, 054408 (2007); S. Zhang et al. *Phys. Rev. Lett.*, **108**, 137203 (2012); S. Z. Wu et al. *Sci. Rep.*, **5**, 8905 (2015); J.-M. Hu et al. *Adv. Mater.*, **28**, 15 (2016); B. Peng et al. *ACS Nano*, **11**, 4337 (2017); W. Hu et al. *Nat. Nanotech.*, **14**, 668 (2019); X. Chen et al. *Nat. Mater.*, **18**, 931 (2019)] in the thin films but also can reversibly tune the light and ultrasound emissions [Y. Zhang et al. *Adv. Mater.*, **24**, 1729 (2012); Y. Zhang et al. *Adv. Mater.*, **30**, 1707007 (2018); J. Yang et al. *J. Semicond.*, **41**, 011901 (2020)].

We have added the information about the PMN-PT substrates in the supplementary information as “Supplementary Note 2. Ferroelectric substrates”.

“0.7PbMg_{1/3}Nb_{2/3}O₃-0.3PbTiO₃ (PMN-PT) is a type of ferroelectric oxide formed by the solid solution of two perovskite ferroelectrics, namely the relaxor ferroelectric PbMg_{1/3}Nb_{2/3}O₃ (PMN) and the ordinary ferroelectric PbTiO₃ (PT). PMN-PT possesses a pseudo-cubic perovskite structure with lattice constants of $a = b = c = 4.02\text{\AA}$ [B. Noheda et al. *Phys. Rev. B*, **66**, 054104 (2002)]. Furthermore, PMN-PT single crystals exhibit excellent ferroelectric, piezoelectric and dielectric properties. For instance, it demonstrates a piezoelectric constant d_{33} up to 2500 pC/N, remanent polarization strength P_r larger than 50 $\mu\text{C}/\text{cm}^2$, and dielectric constant up to 38000 [B. Fang et al. *Eur. Phys. J. Appl. Phys.*, **57**, 30101 (2012)].

When in an unpolarized state, the electric dipoles within the PMN-PT single crystal are randomly oriented. With an increasing electric field applied to the crystal, these initially disordered dipoles tend to align along the direction of the field, inducing a contraction of the crystal lattice perpendicular to the electric field and an expansion parallel to the electric field. Upon application of a bipolar electric field exceeding the coercive field strength, the strain within the PMN-PT single crystal exhibits a characteristic butterfly-shaped response [M. Xu et al. *Acta Phys. Sin.*, **67**, 157506 (2018)].

On the other hand, piezoelectric strain in PMN-PT ferroelectrics originates from non-180° ferroelastic switching, there are multiple switching paths (180°/109°/71°) for ferroelectric polarization upon reversing the electric fields. That is, there could be inequivalent switching possibilities for different switching paths for positive and negative electric fields. Hence, there is a residual strain after the electric field is removed, manifesting as an asymmetric butterfly-shape strain curve, which can be used to achieve non-volatile modulation [S. Zhang et al. *Phys. Rev. Lett.*, **108**, 137203 (2012); S. Wu et al. *Sci. Rep.*, **5**, 8905 (2015)]. Moreover, applying an electric field of one polarity smaller than its coercivity field, so that the switching on one polarity is not completed, can also create asymmetric strains, and generate non-volatile strain states at zero electric fields [Y. Lee et al. *Nat. Commun.*, **6**, 5959 (2015)].

Commonly, (001)-, (011)- and (111)-oriented PMN-PT single crystal are employed as ferroelectric substrates. In this work, we utilized (001)-oriented PMN-PT single-crystal substrates with an in-plane size of $2.5 \times 5 \text{ mm}^2$ and an out-of-plane thickness of 0.3 mm. Before the deposition of the Mn-Ir thin films, the PMN-PT substrates did not undergo any electric-field excitation.”

4. There should be references to the use of Rutherford Backscattering (RBS) in the analyses of thin films. This is not a particularly common technique and it is important that readers are able to understand its uses and limitations.

Response: We fully agree with the Reviewer. The RBS measurement is not a particularly accessible technology. However, the RBS is accurate and widely used to determine the composition of thin film samples. We have added a short discuss and some references about RBS in the analyses of thin films.

“The composition of the Mn-Ir thin film was examined by the Rutherford backscattering (RBS) technology, which is not particularly accessible, but it is accurate and widely used to determine the composition of thin films³³⁻³⁶.”

5. The authors present some very nice TEM results but without full experimental detail – how were the samples prepared, I presume a cut out lamella?

Response: The experimental details of each measurement are described in the part of methods between the main text and the reference in the manuscript, as are TEM measurements. We prepared the TEM samples using the focused ion beam technique.

6. There are no XRD data. This seems an obvious and significant omission since typically XRD, and possibly XRR, are the first structural measurements done to characterize thin films.

Response: We fully agree with the Reviewer that XRD are commonly the first structural measurement for characterizing thin films. Honestly speaking, we have failed to observe thin-film peaks in XRD measurements.

As seen from the TEM images, the Mn-Ir films are polycrystalline and possess small grains. As a result, it has been challenging for us to observe macroscopic diffraction peaks for the XRD measurements. For example, the four-circle X-ray diffraction pattern of a Mn-Ir/PMN-PT film is shown above in Fig. R4. Only the PMN-PT peaks are seen while no thin-film peaks could be observed. One could detect XRD peaks for polycrystalline films when the films exhibit strongly preferred out-of-plane orientations. Otherwise, if the grains are relatively randomly distributed or the grain size is too small, it is difficult to obtain XRD peaks experimentally. Even so, we performed a series of TEM measurements both of the initial state and the high- and low-resistance states of the Mn-Ir thin film, which indicate the crystallinity of our films within local nanoscale regions is excellent due to the high deposition temperature.

Fig. R4 Room-temperature X-ray diffraction pattern of a Mn-Ir/PMN-PT heterostructure.

In addition, we successfully performed the XRR measurement of the Mn-Ir thin film (Fig. R5). Via fitting the XRR pattern, we obtained that the thickness of the Mn-Ir thin film is 26 nm.

Fig. R5 X-ray reflection curve for the Mn-Ir thin film used in the magnetotransport measurements.

7. The film thickness is not given. This is important as the strain-driven changes to magnetic ordering can occur only within a finite distance of the interface. Do the authors have any estimate for the distance over which the strain-mediate effects occur?

Response: The thickness of the Mn-Ir thin films is 26 nm, which was obtained via TEM images and testified by fitting the XRR curve (Fig. R4). Actually, the piezoelectric strain induced by electric fields in the PMN-PT substrates could affect thin films of μm , which is clearly sufficient for our Mn-Ir thin films. For instance, in the previous work, a 50-nm-thickness Mn_3Sn thin film, inserting with a 100-nm-thickness LaAlO_3 buffer layer between PMN-PT substrates, could be modulated by the electric-field-induced piezoelectric strain. [Wang et al. *Acta Mater.* **181**, 537 (2019)]

8. When applying +1.5 kV/cm to the sample with CoFe the resistance state is low, the spins are collinear, and the exchange bias is 10 mT. On applying -4.0 kV/cm the resistance state is high, the spins are non-collinear, and the exchange bias increases to 20 mT. Naively, I would have guessed that the exchange bias is more significant for the case of collinear AF, where there is no field-induced spin canting. Do the authors have any intuition as to why the non-collinear AF exhibits a larger exchange bias effect?

Response: At first glance, it might be thought that the exchange bias effect between the ferromagnetic material and the collinear MnIr phase would be stronger than that between the noncollinear Mn_3Ir phase, but the opposite is true. The exchange bias is more significant for the case of noncollinear Mn_3Ir , compared to the case of collinear MnIr. It has been reported by many works [K. Hoshino et al. *Jpn. J. Appl. Phys.*, **35**, 607 (1996); A. J. Devasahayam et al. *J. Appl. Phys.*, **83**, 7216 (1998); R. Y. Umetsu et al. *Trans. Magn. Soc. Japan*, **3**, 59 (2003); K.-i. Imakita et al. *J. Appl. Phys.*, **97**, 10K106 (2005)] and this is also why Mn_3Ir is commonly used in spin valves to pin the ferromagnetic layer [H. N. Fuke et al. *J. Appl. Phys.*, **81**, 4004 (1997); A. J. Devasahayam et al. *IEEE Trans. Magn.*, **35**, 2 (1999); P. P. Freitas et al. *J. Phys.: Condens. Matter*, **19**, 165221 (2007); C. Y. You et al. *Appl. Phys. Lett.*, **93**, 012501 (2008); B. G. Park et al. *Nat. Mater.*, **10**, 347 (2011)]. The reason why noncollinear Mn_3Ir has large exchange bias could be

attribute to that the triangular antiferromagnetic order causes strong coupling to the magnetic moments in the ferromagnetic layer and the symmetry breaking at the noncollinear Mn₃Ir interface leads to large anisotropy effects [A. Kohn et al. *Sci. Rep.*, **3**, 2412 (2013)].

9. The AHE measurements show the coercivity of the spin-canted L1₂ noncollinear phase to be substantially higher than the magnetometry data of Fig. 1f (>5 T compared to ~2 T). What is the reason for this difference?

Response: The magnetization curve of the Mn-Ir thin film in Fig. 1f just exhibits a minor loop because of the limit of applied external magnetic fields. When a magnetic system is in a cyclic field where the maximum magnetic field is not sufficient to saturate the system, a minor loop will be observed. In the Mn-Ir thin films, the net moments of the Mn atoms are extremely small, on the order of $m\mu_B$. So, we can only measure its magnetization using superconducting quantum interference device (SQUID), a technology with ultrahigh sensitivity to detect small magnetic signal. However, commercial SQUID, such as Quantum Design MPMS used in this work, could only produce a relatively small magnetic field up to 7 T. So, the Mn-Ir thin film is not sufficiently saturated under a magnetic field of 7 T and thus shows a minor loop.

10. Figure 4 needs to have a legend.

Response: Great thanks for Reviewer's careful examination of our manuscript. We have added a legend to each curve in Fig. 4a, c.

11. The TEM images are a particularly nice demonstration of the structural phase change. Do they show any variations with distance from the PMN-PT substrate, either in tetragonality or in the magnitude of the strain? The two regions sampled to calculate the lattice parameters are at different distances from the substrate; it is important to address the degree to which this may be responsible for the observed differences.

Response: We fully agree that there is commonly lattice relaxation in the films, which is dependent on the distance from the substrates. However, we believe that even if lattice relaxation is present, it will not have an observable effect on our TEM measurements. To be specific, the lattice relaxation in a thin film had been determined after the deposition and anneal processes. Moreover, the lattice relaxation in one film sample should be the same. The two TEM samples in Fig. 5 come from exactly two parts of one thin film sample, one under a high resistance state and the other under a low resistance state.

On the other hand, the lattice constants of the PMN-PT substrates are $a = b = c = 4.02 \text{ \AA}$, which are larger than the lattice constants of both the L1₀-collinear phase and L1₂-noncollinear phase of Mn-Ir. Thus, the interface strain between the Mn-Ir thin films and the PMN-PT substrates is tensile strain and the lattice relaxation in the thin films appears as the lattice constant decreases with distance from the substrates until complete relaxation. If the difference in lattice constants between high- and low- resistance states is caused by the interface strain and lattice relaxation, the lattice constants in the region far from the substrate will be smaller than that in the region near the substrate. However, the TEM images show the opposite, the lattice constants in the region far from the substrate (Fig. 5d) are much larger than that in the region near the substrate (Fig. 5b). Therefore, the effect of the interface strain and lattice relaxation is negligible. The variation of the lattice constants and thus the phase change are originated from the piezoelectric strain.

12. What is the dark contrast region between the Mn-Ir and the highly crystalline PMN-PT substrate seen in the TEM images?

Response: As mentioned above and the method section, the TEM samples were prepared by the focused ion beam technique. The ion beam was using Ga ion. When the sample was thinned with a focused ion beam, some Ga ions entered and gathered at the interface between the film and the substrate. Ga has a smaller atomic number than Ir, so it appears as dark contrast region in the HAADF image.

13. The dimensions used of the electrical measurements should be given – the substrate size, probe spacing etc. Were wires directly bonded to the Mn-Ir film? Was the same setup used for both the Mn-Ir and the Mn-Ir/CoFe samples?

Response: The size of the substrates used for the thin-film fabrication and the electrical measurements is $5 \times 2.5 \text{ mm}^2$. The standard four-probe measurement geometry was used for the longitudinal resistance measurements. The contacts were performed by a wire bonding with Al wires. The wires directly bonded to the surface of the Mn-Ir thin films and each probe is spaced about $\sim 1 \text{ mm}$ apart. There are also two lines, one leading from the surface of the Mn film and one leading from the bottom electrode, for connecting the voltage source to apply the gate voltage. The Mn-Ir/CoFe samples used the same setup as the Mn-Ir samples.

14. The data in fig.6a are described as “...originates from the dirty transport feature...”. This is incomprehensible, I have no idea what “dirty transport feature” means.

Response: We sincerely apologize for our confusing statement. The “dirty” means impure, a graphical way of saying that the electrical transport properties of a material are affected by several different factors. It was initially used to describe the superconductors [P. W. Anderson *J. Phys. Chem. Solids*, **11**, 26 (1959); P. W. Brouwer et al. *Phys. Rev. Lett.*, **85**, 5 (2000); Y. V. Fominov et al. *Phys. Rev. B*, **63**, 094518 (2001); A. Gurevich *Phys. Rev. B*, **67**, 184515 (2003)] that have additional defects or impurities and then has been extended to other materials [W. Belzig et al. *Phys. Rev. B*, **54**, 13 (1996); C. Nayak et al. *Phys. Rev. B*, **68**, 104423 (2003); A. A. Istratov et al. *Mater. Sci. Eng. B*, **134**, 282 (2006); J. Linder et al. *Phys. Rev. B*, **76**, 214508 (2007)]. In this work, there could be both electrons and holes contributing to the transport and various scattering in the Mn-Ir system. To make it more accurate, we have revised our description.

“This excellent temperature stability of the resistive states originates from the transport feature that could be impacted by multiple mechanisms in the Mn-Ir system”

15. The data in Fig. 6c shows that the resistance states are stable over a long period of time (6 months). Was the resistance state sampled at several times within this period, or just at the start and the end? The line on the figure appears to have some noise on it, suggesting a large number of measurements were made.

Response: The data of the high- and low-resistance states in the Fig. 6c was collected daily for six months. As the Reviewer mentioned, because the fluctuation is extremely small, it looks like a straight line from the start point to the end point. Actually, there are enormous data points.

16. The calculations in the supplementary information show that the spin reorientation can occur without atomic rearrangement (e.g. due to dislocation formation with strain), a particularly nice

result given that the write cycle endurance in conventional phase change memories is typically limited by atomic migration characteristics. The main text could go further to highlight this point, I think.

Response: That is a nice point. Great thanks for the positive comments on our theoretical work and the opinion which help us a lot to improve our manuscript. We have added a discussion about the write cycle endurance in the main text.

“Besides, the write cycle endurance in conventional phase change memories is typically limited by atomic migration characteristics⁴⁶. However, according to the theoretical simulations, there is no atomic rearrangement during the phase change in the Mn-Ir spin phase change memory, which imply an unlimited write cycle endurance compared to conventional phase change memories.”

17. The supplementary information consists of a series of figures with no commentary or experimental explanations. This is unsatisfactory and must be addressed in any resubmission.

Response: We thank the Reviewer for the kind remaining. We have carefully examined the manuscript to ensure that all supplementary figures in the supplementary information are cited and explained in the main text.

Reviewers' Comments:

Reviewer #1:

Remarks to the Author:

While this reviewer stills has questions about this work the authors adequately responded. As such this review believes all his concerns have been reasonably addressed to warrant moving forward with this manuscript.

Reviewer #2:

Remarks to the Author:

In this revision, the authors have convincingly addressed the comments of the Referees. I appreciate their efforts and recommend the publication of this work on Nat. Commun.

Response to the Reviewers' Comments

Reviewer #1

While this reviewer stills has questions about this work the authors adequately responded. As such this review believes all his concerns have been reasonably addressed to warrant moving forward with this manuscript.

Response: We sincerely thank the Reviewer for this positive response to our revised manuscript.

Reviewer #2

In this revision, the authors have convincingly addressed the comments of the Referees. I appreciate their efforts and recommend the publication of this work on Nat. Commun.

Response: We would like to express our heartfelt gratitude to the Reviewer for her/his positive comments to the previous revisions.

Overall, we are sincerely grateful to all Reviewers for their invaluable inputs and suggestions, with which our work has been greatly improved.